# Abundance estimation for line transect sampling: A comparison of distance sampling and spatial capture-recapture models

Nathan J. Crum⬤*, Lisa C. Neyman⬤, Timothy A. Gowan

Fish and Wildlife Research Institute, Florida Fish and Wildlife Conservation Commission, St Petersburg, Florida, United States of America

* nathan.crum@myfwc.com

## Abstract

Accurate and precise abundance estimation is vital for informed wildlife conservation and management decision-making. Line transect surveys are a common sampling approach for abundance estimation. Distance sampling is often used to estimate abundance from line transect survey data; however, search encounter spatial capture-recapture can also be used when individuals in the population of interest are identifiable. The search encounter spatial capture-recapture model has rarely been applied, and its performance has not been compared to that of distance sampling. We analyzed simulated datasets to compare the performance of distance sampling and spatial capture-recapture abundance estimators. Additionally, we estimated the abundance of North Atlantic right whales in the southeastern United States with two formulations of each model and compared the estimates. Spatial capture-recapture abundance estimates had lower root mean squared error than distance sampling estimates. Spatial capture-recapture 95% credible intervals for abundance had nominal coverage, i.e., contained the simulating value for abundance in 95% of simulations, whereas distance sampling credible intervals had below nominal coverage. Moreover, North Atlantic right whale abundance estimates from distance sampling models were more sensitive to model specification compared to spatial capture-recapture estimates. When estimating abundance from line transect data, researchers should consider using search encounter spatial capture-recapture when individuals in the population of interest are identifiable, when line transects are surveyed over multiple occasions, when there is imperfect detection of individuals located on the line transect, and when it is safe to assume the population of interest is closed demographically. When line transects are surveyed over multiple occasions, researchers should be aware that individual space use may induce spatial autocorrelation in counts across transects. This is not accounted for in common distance sampling estimators and leads to overly precise abundance estimates.

**Data Availability Statement:** All relevant data are within the paper and its Supporting Information files.

**Funding:** This study was supported by the National Oceanic and Atmospheric Administration (NOAA; grant nos. NA14OAR4170108 and NA16NMF4720319). The funders had no role in study design, data collection and analysis, decision to publish, or preparation of the manuscript.

**Competing interests:** The authors have declared that no competing interests exist.

## Introduction

The cornerstone of effective wildlife population management is the ability to estimate population abundance accurately and precisely. Two widely used methods for estimating abundance are distance sampling (DS) and spatial capture-recapture (SCR), a recent development of mark-recapture [1], and there are situations in which both methods could be suitable. Although studies have compared abundance estimates of DS and mark-recapture models [2, 3] and SCR and mark-recapture models [e.g. 4], and demonstrated how these models are related [5], to the best of our knowledge, DS and SCR model performances have not been compared. We compare DS and SCR abundance estimation herein after providing a brief introduction for both approaches. For in-depth descriptions we refer interested readers to Buckland et al. [6] for DS and Royle et al. [7] for SCR.

During a DS survey, one or more observers conduct at least one search for individuals of a species of interest. When observers detect an individual, they record the distance between themselves and the detected individual. Distances to detected individuals are used to estimate the scale of a detection function, which describes the probability that an individual is detected, given its distance from the observer. The detection function and abundance can be estimated simultaneously as in one-stage, hierarchical distance sampling [8]. Alternatively, a two-stage approach can be taken where the detection function is estimated first, and abundance second using a Horvitz-Thompson estimator based on the average detection probability over the surveyed region and the number of individuals detected [6].

There are two primary DS survey types, line transect and point transect [1, 6]. While both methods are useful, depending on the species or object of interest, we focus solely on line transect surveys, which involve observers traveling along a transect line and recording the perpendicular distance between detected individuals and the line. Individuals in the population of interest need not be individually identifiable for DS, although it is assumed that an individual is seen at most once over a transect line.

A key assumption underlying DS is certain detection on the transect line [6]. However, detection is often imperfect on the transect line due to availability and perception biases. For instance, some individuals are unavailable for detection or observers do not detect all individuals that are located on the transect line and are available for detection, respectively. When either bias is present, survey or analytical methods can be modified to relax the assumption of certain detection on the transect line and reduce the negative bias of resulting abundance estimates [9]. One such approach to estimate perception bias is to use a double-observer design, or mark-recapture distance sampling (MRDS), which involves two independent observers scanning the survey area [6, 9, 10]. The proportion of detections made by one observer to detections made by both observers is used to estimate the probability of detection given an individual is located on the transect line [6, 11]. Alternatively, a separate study may be conducted to estimate detection probability on the transect line for the species of interest [12–15].

SCR, unlike DS, requires that individuals in the population of interest be individually identifiable. Identifying marks may be natural, such as humpback whale fluke or tiger stripe patterns [16, 17], a DNA sample collected from hair or scat [18, 19], or human imposed, such as leg bands on birds [20]. A surveyor attempts to capture or detect individuals, often using an array of traps or detectors, from the population of interest. When individuals can be detected at most once during a sampling occasion, multiple sampling occasions are needed for a SCR analysis. The surveyor records encounter histories for each individual, denoting the occasions during which each individual was and was not detected. Additionally, the location of each detector and each detection is recorded for SCR analyses. Encounter histories can be used to

estimate detection probability, which, along with the number of individuals detected at least once, informs abundance estimation.

SCR was developed to address two shortcomings of traditional mark-recapture [7, 21, 22]. First, abundance estimates from mark-recapture are not associated with a well-defined study area, so population density cannot be formally estimated. Second, mark-recapture does not account for individual heterogeneity in detection probability due to individuals' locations relative to detectors. SCR addresses these issues by incorporating the spatial data inherent to an individual's encounter history, i.e. where they were detected and where they were not. Specifically, each individual is associated with an unobservable activity center, which represents the center of an individual's movement or average location across sampling occasions. SCR estimates the density of activity centers across a spatial state-space defined by the researcher. The probability that an individual is detected at a detector on an occasion is modeled as a function of the distance between the individual's activity center and the detector. This detection function is analogous to the detection function used in DS.

SCR models can also accommodate data collected through search encounter field methods where individuals are identifiable [22]. These field methods include surveying points, plots, or along lines, such as in DS surveys. Search encounter SCR models differ from traditional SCR models in that individuals can be detected anywhere in the state space, not just at the location of detectors. However, to the best of our knowledge, all applications of the search encounter SCR model treat surveys as a trapping grid, where survey effort and the locations of detections are aggregated into a grid of cells [e.g., 23–25]. If survey effort occurs in a cell, then the "trap" at the center of the cell is considered active, i.e., individuals can be detected there. The amount of survey effort expended within each cell can be used as a covariate on detection probability. However, when detections can occur away from a transect line, as in DS surveys, an individual may be detected in cells other than the cell containing the line transect from which the individual was detected. Therefore, this approach presents a challenge to modeling survey effort when detections occur away from the line. The search encounter SCR model proposed by Royle et al. [22], which has yet to be broadly applied, avoids this challenge. It models detection probability as a function of the distance between an individual's location and the location of survey effort, rather than aggregating survey effort into a trapping grid. Like DS models for multiple occasions, SCR assumes population closure (no births, deaths, immigration, or emigration) across occasions and, like all mark-recapture models, that heterogeneity in detection probability across individuals is modeled. Unlike DS, search encounter SCR models do not require certain detection along the transect line and can model the space use of individuals across occasions.

One example of a species whose abundance is often estimated with DS but could be estimated with SCR is the endangered North Atlantic right whale (NARW) *Eubalaena glacialis* [26, 27]. Aerial line transect surveys are conducted off the southeastern US coast (SEUS) on most days with suitable weather each winter (December through March) to monitor the NARW population while on their winter calving grounds [28, 29]. Each NARW is uniquely identifiable by callosity patterns on its head [30], and survey protocol dictates that planes break track to photograph and identify every detected individual and obtain their exact location [29]. Although this survey protocol of multiple sampling occasions and identifying individuals is not needed for DS, this combination produces data that can be modeled with both DS and SCR.

Conceptually, search encounter SCR may be more appropriate for use in estimating NARW abundance than DS. NARWs are not always available for detection during a survey (i.e., are at a depth at which they cannot be seen) and may be missed by observers even when they are located along the transect line and are available to be detected. SCR estimates the

probability of detecting an individual located on the transect line based on information from individual identifications, while DS requires additional data to be collected to correct cases with imperfect detection on the transect line. Moreover, NARWs often move between sampling occasions, and inference about NARW movement may be of interest. SCR explicitly models individuals' space use across sampling occasions, while DS does not.

However, the performance of the search encounter SCR model has not been compared to DS. Given the lack of understanding around how the increased flexibility of SCR models may influence abundance estimation, our objective was to compare the accuracy and precision of abundance and other parameter estimates from DS and SCR models. We analyzed simulated datasets and NARW data from January 2010 with DS and SCR models for this comparison.

## Material and methods

### Simulation scenarios

We analyzed simulated line transect datasets with DS and SCR models to compare their performance. For each dataset, we simulated a population of individuals where each individual was associated with an activity center, $\mathbf{s} = (s_x, s_y)$, located within the state space (study area). Individuals moved about these activity centers, resulting in locations or movement outcomes, $\mathbf{u} = (u_x, u_y)$, during each occasion of the simulation. We overlaid transect lines on the state space, and the probability that an individual was detected on an occasion was a function of the individual's distance to the closest point on a transect line. Each transect line was surveyed during each occasion in these simulations, although this is not a requirement for either DS or SCR. From each simulation we generated capture histories that included each individual's location on the occasions it was detected.

We simulated datasets based on four values of the probability of detection on the transect line, commonly referred to as g(0), g(0) $\epsilon$ (1, 0.8, 0.5, 0.3), because our SCR model allows estimation of g(0) without additional data, while DS models require additional data to be collected to estimate g(0). These additional data often come in the form of having an additional observer during surveys (e.g. MRDS) or by estimating g(0) independently of the line transect surveys. Therefore, we simulated datasets generated from single- and double-observer sampling designs for each value of g(0) less than one, resulting in seven scenarios (single- and double-observer designs for g(0) $\epsilon$ (0.8, 0.5, 0.3) and a single-observer design for g(0) = 1).

We simulated 1,000 datasets for each scenario with each dataset having a population size of 100 individuals. The number of occasions varied across different scenarios, so that the average number of detections was similar across scenarios (between 50 and 70). Activity centers were uniformly distributed across a state space ($0 \leq x \leq 40$, $0 \leq y \leq 40$; Fig 1), where we denote the activity center of individual $i$: $(s_{i,x}, s_{i,y})$.

Individuals' locations on each occasion were distributed according to a bivariate normal distribution around their respective activity centers, with movement scale parameter, $\sigma_m = 2.5$, such that the probability of the location of individual $i$ on occasion $j$, $\mathbf{u_{i,j}}$, occurring within some area of space, $\mathbf{A}$, such as a grid cell in a gridded state space, is a function of the distance, $d_{movement,i,j}$, between the individual's location and its activity center, following:

$$P\left(\mathbf{u_{i,j}} \in \mathbf{A}\right) = \int_A \frac{1}{2\pi\sigma_m^2} e^{-\frac{1}{2}\left(\frac{d_{movement,i,j}^2}{\sigma_m^2}\right)} d\mathbf{A} \qquad (1)$$

Ten horizontal line transects were surveyed each occasion, although each line need not be surveyed every occasion in practice (Fig 1). Line transects were defined as y $\epsilon$ (11, 13, 15, 17, 19, 21, 23, 25, 27, 29), with endpoints at x = 10 and 30. The probability that individual $i$ was

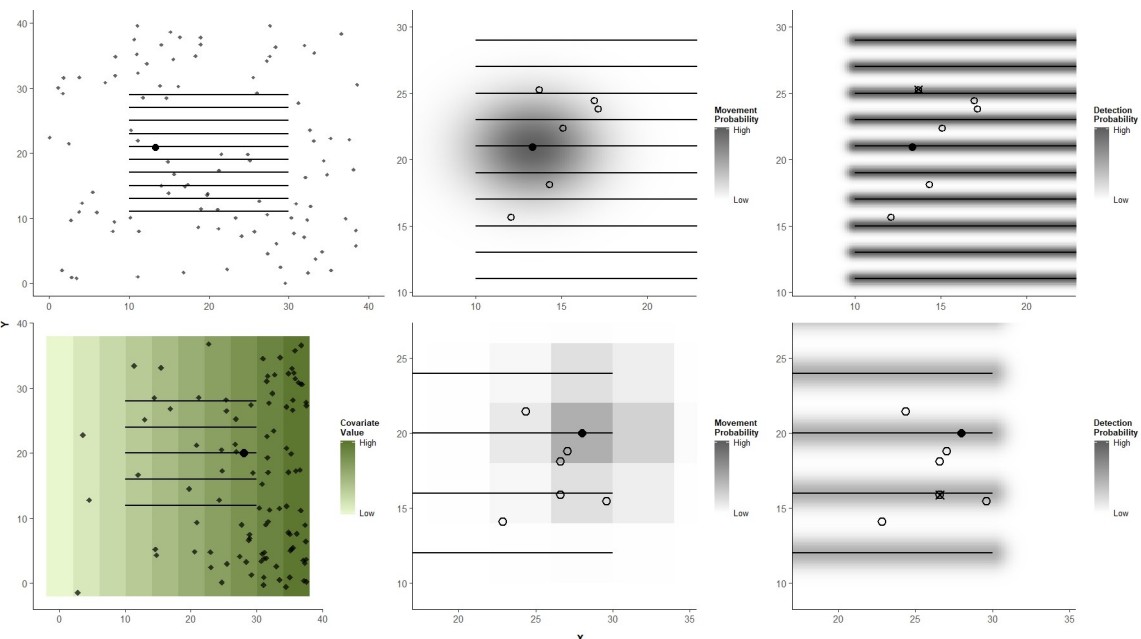

**Fig 1. Simulated datasets.** Examples of simulated datasets following a spatial capture-recapture model where density was uniform across the state space (top row) or a function of a spatial covariate in the state space (bottom row). Each individual was associated with an activity center, depicted by the points in the left column. Individuals were located around their respective activity centers on each sampling occasion, represented for one individual by the open circles in the middle column. The probability of an individual moving to a particular location during an occasion was a function of the distance from its activity center, with the individual being more likely to be located close to rather than far from the activity center. These locations were also distributed as a function of the spatial covariate in the spatial covariate scenario (bottom row). Line transect surveys, represented by the horizontal lines, were conducted during each sampling occasion. The probability that an individual was detected on an occasion was a function of the distance between its location on that occasion and the closest point on a transect line, with the probability being higher the closer an individual was to a line (right column). The locations of detections are depicted by the crossed-out circles in the right column.

detected on occasion $j$, denoted as $y_{i,j} = 1$, was a function of g(0) and the distance between the individual and the closest transect on occasion $j$, $d_{detection,i,j}$, with a scale parameter, $\sigma_d$, of 1/3.

$$P(y_{i,j} = 1) = g(0) \left( e^{-\frac{d^2_{detection,i,j}}{2\sigma^2_d}} \right) \qquad (2)$$

We chose the above sampling configuration and values for $\sigma_m$ and $\sigma_d$ such that individuals with activity centers located at the edge of the state space would have a negligible probability of being detected over the course of sampling. In single-observer scenarios, we simulated whether an individual was detected or not on an occasion from a Bernoulli distribution. In double-observer scenarios, we simulated the number of observers that detected an individual on an occasion from a binomial distribution with two draws.

We were also interested in comparing the performance of DS and SCR models when estimating how density is associated with a spatial covariate. We simulated 250 datasets with 150 individuals, 10 sampling occasions, and g(0) = 0.8 for this spatial covariate scenario. We used the same state space as in previous scenarios but discretized it into G = 100 4-by-4-unit grid cells, where each cell was associated with a covariate value (Fig 1). Activity centers were distributed across cells of the state space according to a multinomial distribution, with the probability of the activity center of individual $i$, $s_i$, occurring in cell $k$ being a log-linear function of the

spatial covariates.

$$P(\mathbf{s_i} \in \text{cell}_k) = \frac{e^{\beta_0 + \beta_1 \text{Covariate}_k}}{\sum_{n=1}^{G} e^{\beta_0 + \beta_1 \text{Covariate}_n}} \tag{3}$$

Here, we set the values for parameters $\beta_0$ and $\beta_1$ such that the expected population size, $\sum_{n=1}^{G} e^{\beta_0 + \beta_1 \text{Covariate}_n}$, was 150.

Individuals' locations were modeled on two spatial scales. The coarser scale was the cell in which an individual was located on an occasion, which followed a multinomial distribution, with the probability of being located in cell $k$ being a function of that cell's distance from the individual's activity center, with a scale parameter $\sigma_m = 4$, and the cell's covariate value [31].

$$P(\mathbf{u_{i,j}} \in \text{cell}_k) = \frac{e^{\frac{-d_k^2}{2\sigma_m^2} + \beta_1 \text{Covariate}_k}}{\sum_{n=1}^{\# \text{ of cells}} e^{\frac{-d_n^2}{2\sigma_m^2} + \beta_1 \text{Covariate}_n}} \tag{4}$$

Here we used the same $\beta_1$ parameter describing the effect of the covariate from the activity center distribution, although SCR models can model these effects separately. Then, for the finer spatial scale, the individual's location within a cell followed a uniform distribution.

For this spatial covariate scenario, five single-observer line transects were surveyed each occasion (Fig 1). Line transects were defined as $y \in (12, 16, 20, 24, 28)$ with endpoints $x = 10$ and 30. The probability that an individual was detected follows from Eq 2, with a detection scale parameter, $\sigma_d$, of 2/3.

## Distance sampling

We formulated our line transect DS models following Kery and Royle [32], who describe hierarchical, one-stage DS models. We used a Bayesian approach with data augmentation to estimate total abundance within the state space (N), abundance within the sampled region ($N_{sampled}$), the detection scale ($\sigma_d$), and the probability of detection at zero distance from the survey line (g(0)) in double observer, MRDS scenarios. $N_{total}$ was derived from the estimate of $N_{sampled}$ following:

$$N = N_{sampled} * \frac{\text{Area}_{\text{state space}}}{\text{Area}_{\text{sampled}}} \tag{5}$$

where $\text{Area}_{\text{state space}}$ is the area of the state space. The area sampled, $\text{Area}_{\text{sampled}}$, is based on the cumulative length of transect lines surveyed across sampling occasions and the truncation distance, which is described below.

Data augmentation is performed by adding M—n latent detection distances to the model, where M is the largest population size permitted by the model and n is the number of detections. Abundance within the sampled region followed a binomial distribution:

$$N_{sampled} \sim \text{Binomial}(M, \psi)$$

where $\psi$ is the data augmentation parameter used to estimate the proportion of the M potential individuals that were truly part of the population [32]. In uniform density scenarios $\psi$ followed a uniform distribution:

$$\psi \sim \text{Uniform}(0, 1)$$

In scenarios where density was modeled as a function of a spatial covariate, $\psi$ was a function of

expected abundances across grid cells:

$$\psi = \frac{\sum_{k=1}^{K} \lambda_k}{M} \tag{6}$$

$$\lambda_k = e^{\beta_0 + \beta_1 \text{Covariate}_k} \tag{7}$$

where $\lambda_k$ and $\text{Covariate}_k$ are the expected abundance and covariate value of cell k, respectively, and $\beta_0$ and $\beta_1$ are estimated parameters. Additionally, in the spatial covariate scenario the cell in which an individual was located followed a categorical distribution:

$$u_i \sim \text{Categorical}\left(\frac{\lambda_1}{\sum_{k=1}^{K} \lambda_k}, \frac{\lambda_2}{\sum_{k=1}^{K} \lambda_k}, \ldots, \frac{\lambda_K}{\sum_{k=1}^{K} \lambda_k}\right)$$

where $u_i$ is the cell in which individual i is located.

For line transect data the distances between individuals and transect lines follow a uniform distribution:

$$d_i \sim \text{Uniform}(0, d_{\text{truncation}})$$

where $d_i$ is the distance from individual i to the transect line and $d_{\text{truncation}}$ is the truncation distance, which defines the boundary of the sampled region around a transect. In our analyses the truncation distance was half the distance between adjacent transect lines. However, our simulations and the NARW application allowed for detections beyond the endpoints of transect lines, which is atypical for line transect distance sampling. Distances between individuals and the endpoints of transect lines follow a triangular distribution, as in point transect distance sampling [33]:

$$d_i \sim \text{Triangular}(0, d_{\text{truncation}})$$

Additional details describing the simultaneous analysis of detection distances within and outside transect line endpoints can be found in the S3 Appendix. Finally, the number of times individual i was detected or not, $y_i$, followed a binomial distribution:

$$y_i \sim \text{Binomial}\left(g(0)e^{-\frac{d_i^2}{2\sigma_d^2}}, X\right)$$

where x, the number of draws, was one for single observer models and two for double observer models. In double observer models we estimated g(0) from the data with a non-informative uniform prior distribution, whereas in single observer models g(0) was either fixed at the simulating value of g(0) or drawn from an informative beta prior distribution. Informative beta prior distributions had a mean equal to the value of g(0) used to simulate the dataset being analyzed and a standard deviation of 0.1 in uniform density models and 0.05 in models with spatial covariates. Use of informative priors was intended to replicate the approach of using independent data to estimate g(0). In all models we estimated $\sigma_d$ from the data with a non-informative uniform prior distribution.

## Spatial capture-recapture

We formulated our SCR models following Royle et al. [22]. We again used a Bayesian approach with data augmentation to estimate total abundance within the state space (N), the movement scale ($\sigma_m$), the detection scale ($\sigma_d$), and the probability of detection at zero distance from the

survey line (g(0)). Abundance within the state space followed a binomial distribution:

$$N \sim \text{Binomial}(M, \psi)$$

As in the distance sampling models, $\psi$ followed a uniform distribution in uniform density models. In models that included a spatial covariate, $\psi$ was a function of expected abundances across grid cells:

$$\psi = \frac{\sum_{k=1}^{K} \lambda_k}{M} \tag{8}$$

$$\lambda_k = e^{\beta_0 + \beta_1 \text{Covariate}_k} \tag{9}$$

In uniform density models the activity center of individual i, $\mathbf{s_i}$, followed a uniform distribution:

$$\mathbf{s_i} \sim \text{Uniform}(S)$$

where S is the state space. In models with a spatial covariate, the grid cell in which $\mathbf{s_i}$ was located followed a categorical distribution:

$$\mathbf{s_i} \sim \text{Categorical}\left( \frac{\lambda_1}{\sum_{k=1}^{K} \lambda_k}, \frac{\lambda_2}{\sum_{k=1}^{K} \lambda_k}, \dots, \frac{\lambda_K}{\sum_{k=1}^{K} \lambda_k} \right)$$

The location of individual i on occasion j, $\mathbf{u_{i,j}}$, followed a bivariate normal distribution in uniform density models:

$$\mathbf{u_{i,j}} \sim \text{Normal}(\mathbf{s_i}, \mathbf{\Sigma})$$

$$\mathbf{\Sigma} = \begin{bmatrix} \sigma_m & 0 \\ 0 & \sigma_m \end{bmatrix} \tag{10}$$

In models with a spatial covariate, the grid cell in which $\mathbf{u_{i,j}}$ was located followed a categorical distribution:

$$\mathbf{u_{i,j}} | \mathbf{s_i} \sim \text{Categorical}\left( \frac{\gamma_1}{\sum_{k=1}^{K} \gamma_k}, \frac{\gamma_2}{\sum_{k=1}^{K} \gamma_k}, \dots, \frac{\gamma_K}{\sum_{k=1}^{K} \gamma_k} \right)$$

$$\gamma_k = e^{\left( -\frac{d_k^2}{\sigma_m^2} + \beta_1 * \text{Covariate}_k \right)} \tag{11}$$

where $d_k$ is the distance from $\mathbf{s_i}$ to the center of cell k. The location of $\mathbf{u_{i,j}}$ within a grid cell followed a uniform distribution. We estimated $\sigma_m$ from the data using a non-informative uniform prior distribution. Whether individual i was detected or not on occasion j, $y_{i,j}$, followed a binomial distribution:

$$y_{i,j} \sim \text{Binomial}\left( g(0)e^{\frac{d_{i,j}^2}{2\sigma_d^2}}, X \right)$$

where $d_{i,j}$ is the distance between $\mathbf{u_{i,j}}$ and the closest transect line and the number of draws, x, is two in double observer designs and one in single observer designs. We estimated g(0) and $\sigma_d$

from the data using non-informative uniform prior distributions. This detection model is identical to that of our distance sampling model and can accommodate continuous line transect survey effort. In contrast, Royle et al. [22] present hazard models for detection and approximated line transect survey effort using a set of evenly spaced points along the transects. Details for calculating the distance between $\mathbf{u_{i,j}}$ and continuous transect lines can be found in the S3 Appendix.

## Analysis of simulated datasets

We used the Nimble package [34] in program R 3.6.2 [35] to write and implement all DS and SCR models [S3 Appendix]. For each simulated dataset we ran three Markov chain Monte Carlo (MCMC) chains of 25,000 iterations for each model. We checked for convergence of MCMC chains using the r-hat statistic [36]. We discarded the fewest of 5,000, 10,000, or 15,000 iterations as burn-in for each simulation while ensuring that r-hat values were less than 1.1 for all parameters. For instance, if all chains converged within the first 5,000 iterations, the first 5,000 iterations were discarded, but if all chains did not converge within the first 5,000 iterations but did within 10,000 iterations, the first 10,000 iterations were discarded, and so on. If the chains of any parameter did not converge by 15,000 iterations, all analyses conducted on that dataset were omitted and a new dataset was simulated and analyzed to replace it. However, we did not simulate new datasets in such cases for the spatial covariate scenario because of the greater runtime required for those models. We report the average posterior mean and 95% credible interval, as well as the 95% credible interval coverage, average root mean squared error (RMSE), and average bias for each parameter in each scenario. The 95% credible interval coverage was calculated as $\sum_{i=2}^{n} I(\theta \geq \hat{\theta}_{i,0.025}) * I(\theta \leq \hat{\theta}_{i,0.975})/n$, where n is the number of simulations, $\theta$ is the true value of a parameter, $\hat{\theta}_{i,0.025}$ and $\hat{\theta}_{i,0.975}$ are the 0.025 and 0.975 quantiles of the posterior distribution of parameter $\theta$ of the i-th simulation, and I is an indicator function that equals one if its argument is true and zero if it is false. Average root mean squared error and bias were calculated as $\sum_{i=1}^{n} \sum_{t=1}^{T} \left( \sqrt{(\hat{\theta}_{i,t} - \theta)^2}/T \right)/n$ and $\sum_{i=1}^{n} \sum_{t=1}^{T} ((\hat{\theta}_{i,t} - \theta)/T)/n$, respectively, where T is the number of MCMC iterations after discarding the burn-in, and $\hat{\theta}_{i,t}$ is the estimate of $\theta$ at the t-th MCMC iteration for the i-th simulated dataset $\hat{\theta}_t$.

## Estimating abundance of North Atlantic right whales

Using single-observer DS and SCR models, we estimated abundance of NARWs in the SEUS from aerial survey data collected between 16 January 2010 and 31 January 2010. Specifically, we estimated abundance over a 76,800 km² state space that ranges from 28.0˚ N to 33.8˚ N and from the coast of Florida, Georgia, and South Carolina to waters 70 m deep (Fig 2). Aerial line transect surveys were flown for 10 of the 16 days. Subsets of 82 transect lines were flown each of these 10 days. The transect lines flown ranged from 29.6˚ N to 33.0˚ N and from shore to as far as 65 km offshore. When a NARW was detected, the aerial survey team broke from the transect line to photograph the NARW for identification purposes and to document the exact location of the NARW. See Gowan and Ortega-Ortiz [29] for a further description of field methods used. Over these 10 survey days 91 individual NARWs were detected (38 on one occasion, 27 twice, 12 three times, 11 four times, and 3 five times).

We present two DS and SCR models of right whale density to demonstrate the application of both models to the same dataset. The first assumes density is constant across space, and the second models spatial variation in density as a quadratic function of depth analogous to our spatial

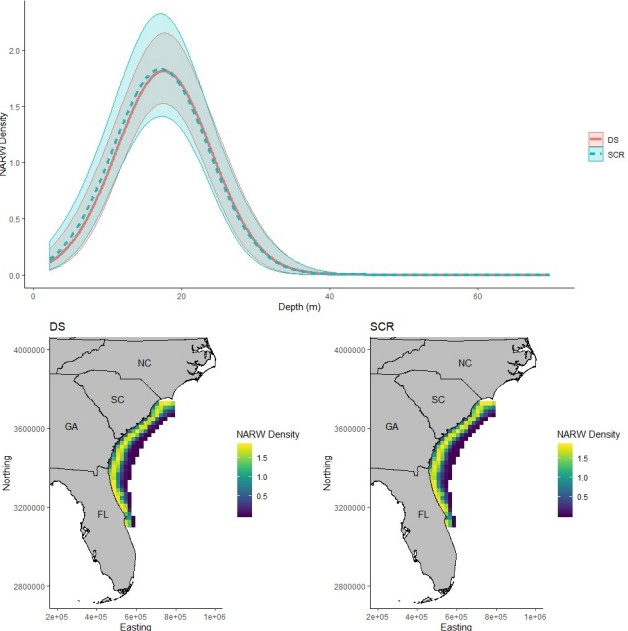

**Fig 2. North Atlantic right whale density.** Expected North Atlantic right whale (NARW) density per 400 km$^2$. NARW density was estimated as a quadratic function of depth within the state space using distance sampling (DS; bottom left panel and red in top panel) and spatial capture-recapture (SCR; bottom right panel and blue in top panel). Posterior mean expected density and 95% credible intervals across the range of depth in the state space is illustrated in the top panel for both DS (red) and SCR (blue) models. Bottom panels illustrate posterior mean expected density across the state space for DS (left) and SCR (right) models that included depth and depth$^2$ as covariates on density.

covariate scenario. Additional models that include different detection functions and density models would be needed to make robust multi-model inference regarding NARW abundance, however such a model set is beyond the scope of this study. In our DS analyses of right whale data we fixed the value of g(0) to 0.6 following the work of Hain et al. [37], who found right whale availability for detection in the Southeast was approximately 0.6. Uncertainty in this estimate of availability was high and including that uncertainty into a prior distribution for g(0) led to slow model convergence and unreasonably large abundance credible intervals in preliminary tests; therefore we did not account for this uncertainty in g(0). We used a Bayesian approach for all models and augmented the encounter histories of the 91 observed individuals with 259 unobserved encounter histories of individuals that may have been part of the population, resulting in a maximum possible abundance of 350. Preliminary analyses indicated that enough unobserved encounter histories were added through data augmentation, as the posterior distribution for abundance was not truncated. We ran three MCMC chains of 20,000 iterations for the uniform density models and three chains of 40,000 iterations for the models that included depth as a covariate on density. We discarded 5,000 iterations as burn-in for each model and report the posterior means and credible intervals of estimated parameters.

## Results

### Simulations

SCR abundance estimates were positively biased, from 1–9%, across all scenarios (however, see S1 Appendix, which shows SCR abundance estimates of larger datasets are unbiased), while the bias of DS abundance estimates was more varied (Table 1). DS models with a fixed g(0)

**Table 1. Simulation summary.**

| g(0) | # Occasions | # Observers | Model | Mean | 95% CI | 95% CI Coverage | RMSE | Bias |
|------|-------------|-------------|-------|------|--------|-----------------|------|------|
| 1 | 4 | 1 | SCR | 104.3 | (73.0, 143.1) | 0.944 | 18.5 | 4.3 |
| | | | DS–fixed g(0) | 102.2 | (72.8, 138.9) | 0.854 | 22.6 | 2.2 |
| 0.8 | 6 | 1 | SCR | 109.1 | (75.4, 152.7) | 0.945 | 20.8 | 9.1 |
| | | | DS–fixed g(0) | 101.3 | (73.4, 134.9) | 0.841 | 22 | 1.3 |
| | | | DS–informed g(0) | 105.3 | (69.9, 162.3) | 0.935 | 23.4 | 5.3 |
| 0.8 | 5 | 2 | SCR | 103.6 | (76.2, 136.8) | 0.956 | 15.6 | 3.6 |
| | | | DS–double observer | 103.9 | (80.3, 134.1) | 0.819 | 19.7 | 3.9 |
| 0.5 | 10 | 1 | SCR | 105.4 | (72.5, 148.6) | 0.953 | 19.2 | 5.4 |
| | | | DS–fixed g(0) | 102.7 | (73.9, 137.6) | 0.866 | 21.3 | 2.7 |
| | | | DS–informed g(0) | 113 | (66.1, 194.9) | 0.98 | 26.8 | 13 |
| 0.5 | 7 | 2 | SCR | 105 | (76.2, 140.6) | 0.952 | 16.2 | 5 |
| | | | DS–double observer | 112.9 | (76.5, 173.6) | 0.872 | 31 | 12.9 |
| 0.3 | 20 | 1 | SCR | 103.9 | (73.7, 142.7) | 0.954 | 17.6 | 3.9 |
| | | | DS–fixed g(0) | 100.6 | (73.8, 132.4) | 0.85 | 20.8 | 0.6 |
| | | | DS–informed g(0) | 130.3 | (58.4, 277.7) | 0.997 | 40.9 | 30.3 |
| 0.3 | 11 | 2 | SCR | 104.7 | (75.5, 141.1) | 0.948 | 17.1 | 4.7 |
| | | | DS–double observer | 125 | (68.8, 233.4) | 0.91 | 53.9 | 25 |
| 0.8 | 10 | 1 | SCR–spatial covariate | 156 | (116.1, 209.2) | 0.965 | 18.1 | 6 |
| | | | DS–fixed g(0)–spatial covariate | 153.4 | (115.1, 200.3) | 0.881 | 20.3 | 3.4 |
| | | | DS–informed g(0)–spatial covariate | 152.7 | (111.6, 204.7) | 0.895 | 19.3 | 2.7 |

Model performance based on simulated datasets for spatial capture-recapture models (SCR) and distance sampling models (DS), where g(0) (detection on the transect line) was either fixed at the true value, informed via a prior distribution, or estimated via a double observer sampling protocol (i.e. mark-recapture distance sampling). The average posterior mean, 95% credible interval (CI), 95% CI coverage, root mean squared error (RMSE), and bias for abundance are reported for each scenario.

were negligibly biased (-1%–2%) across all scenarios, while double-observer and informed g(0) DS models had increasingly positive bias as the true value of g(0) decreased (bias <5% with g(0) = 0.8 and 25–30% with g(0) = 0.3) (Table 1).

RMSE of SCR abundance estimates was lower than that of all DS models in all scenarios (Table 1). DS models with fixed g(0) had RMSE 6–22% higher than SCR models across scenarios. RMSE of abundance estimates from informed g(0) and double-observer DS models increased as the true value of g(0) decreased. Additionally, RMSE of abundance estimates from these models was 12% and 26% higher than that of SCR in scenarios where g(0) was 0.8. When g(0) was 0.3 RMSE was 133 and 215% higher than that of SCR (Table 1).

The estimated abundance credible interval range was 17–25% narrower for fixed g(0) DS models than for SCR models across all scenarios. The credible interval range increased as the true value of g(0) decreased for double-observer and informed g(0) models, increasing 206% and 137%, respectively, from when g(0) was 0.8 to 0.3 The credible interval ranges of these two models were wider than that of SCR in all scenarios except the double-observer scenario with g(0) equal to 0.8 (Table 1).

Abundance 95% credible interval coverage was nominal, i.e. not significantly different from 95%, for SCR models in all scenarios. Coverage was below nominal for fixed g(0) and double-observer DS models in all scenarios. Coverage was below nominal for the informed g(0) DS model when g(0) was 0.8 and was above nominal when g(0) was 0.5 and 0.3 (Table 1).

Both the bias and RMSE of the detection scale estimates were lower for SCR than for all DS models across all scenarios. Likewise, bias and RMSE of the estimated effect of the spatial covariate were lower for SCR than for DS in the spatial covariate scenario. The estimated

movement scale from SCR models was negatively biased across all scenarios. In uniform density scenarios with g(0) equal to 1, 0.8, and 0.5 all top level parameters (N, g(0), $\sigma_d$, and $\sigma_m$) converged for >99% of both SCR and DS models. In the scenario with g(0) equal to 0.3, top level parameters converged in all DS models that fixed the value g(0), in 73.0% of DS models that drew g(0) from an informative prior distribution, and in 79.3% of SCR models. In the spatial covariate scenario convergence was reached for all top level parameters (N, g(0), $\beta_0$, $\beta_1$, $\sigma_d$, and $\sigma_m$) in 64% of fixed g(0) DS analyses, in 60.8% of informed g(0) analyses, and in 79.2% of SCR analyses [S1 Appendix].

### North Atlantic right whale abundance

The uniform density DS model estimated a population size of 247.5 (210–285 95% CI), while the uniform density SCR model estimated a population size of 160.2 (134–193 95% CI). The value of g(0) for the DS model was fixed at 0.6, and the uniform density SCR model estimated a value of g(0) of 0.832 (0.662–0.984 95% CI). The uniform density SCR and DS models estimated a similar $\sigma_d$ of 3.51 km (2.94–4.24 km 95% CI) and 3.47 km (2.91–4.18 km 95% CI), respectively. The SCR model estimated a $\sigma_m$ of 142.6 km (119.3–182.4 km 95% CI).

The SCR and DS models that modeled density as a quadratic function of depth estimated similar population sizes of 159.0 (131–193 95% CI) and 156.4 (133–182 95% CI), respectively. Additionally, both models estimated density to be similarly associated with depth (Fig 2). The SCR model estimated g(0) to be 0.558 (0.441–0.696 95% CI). Again, the SCR and DS models estimated similar values of $\sigma_d$, 3.79 km (3.11–4.67 km 95% CI) and 3.82 km (3.12–4.69 km 95% CI). The SCR model estimated a $\sigma_m$ of 153.9 km (124.0–201.0 km 95% CI).

## Discussion

We compared the accuracy and precision of search encounter spatial capture-recapture (SCR) and hierarchical distance sampling (DS) abundance estimators using simulated data sets and found that the SCR estimator outperformed all DS estimators. The SCR abundance estimator had lower RMSE than all DS estimators and had nominal credible interval coverage, whereas DS estimators had below nominal coverage. Additionally, while the SCR abundance estimator was slightly positively biased, the DS estimators were as or more positively biased when uncertainty in g(0) was modeled. The search encounter SCR model has not been applied beyond its conception [22, 38], and we are unaware of any such comparison between DS and SCR models. However, the search encounter SCR model outperformed DS models in our scenarios, suggesting that future research and use of the search encounter SCR model is warranted.

Assumptions of conventional DS include the following: all animals on the transect line are detected (g(0) = 1), animals do not move in response to the observer, distance to animals is measured without error, and animals are detected independently of one another [6]. Our simulated datasets did not violate these assumptions, other than g(0) ≠ 1, which we accounted for in our modeling approaches. Despite this, the DS abundance estimates were overly precise (i.e. variance was underestimated and credible interval coverage was below nominal) and became more so in scenarios with more sampling occasions [S1 Appendix]. We suspect that this over-precision is due to the DS model underestimating variance by not explicitly accounting for the space use of individuals relative to the spatial configuration of transect lines across sampling occasions. This space use of individuals can cause spatial autocorrelation in the number of detections across transect lines. Models have been developed for repeated DS surveys to account for individuals moving in to and out of the sampled region, i.e., temporary emigration [39]. Still, variance is not appropriately estimated by these models when individuals have different probabilities of temporarily emigrating, as in our simulations where the overlap of an

individual's space use with the sampled region depended on the location of its activity center [39]. Developments in hierarchical DS have been proposed to account for non-independence of detections at discrete sites over repeat sampling occasions [40]. Advancing this idea to cases where the study area is continuous, rather than composed of discrete sites, could address the over-precision we found in DS abundance estimates. Additionally, variance estimation has received more attention for two-stage DS models than the one-stage hierarchical model we applied [41]. However, variance is still underestimated by a two-stage DS model that accounts for repeated sampling occasions, whether variance is estimated analytically or through non-parametric bootstrapping [S2 Appendix]. Moving block bootstrapping [42, 43] and parametric bootstrapping [44] are additional specialized approaches that may provide unbiased variance estimation but were not tested here. We evaluated simple space use models here that may not approximate the space use patterns of many species. Moreover, the space use models of our simulations met the assumptions of the SCR models. Therefore, we did not evaluate whether SCR abundance estimates are robust to any misspecification of the space use model. Additional work would be needed to determine if DS underestimates the variance of abundance and if SCR abundance estimates are robust under other individual space use models.

The DS models required modification to account for the location along the transect line where individuals were detected, i.e. within or outside the transect line's endpoints. Detections within endpoints were generated from line transect sampling, while detections outside endpoints were generated from point transect sampling. This modification is not available in common analytical tools for DS, such as the Distance package [45] in program R, and required a custom sampler built through the Nimble package [33; S3 Appendix]. Researchers using DS to model line transect surveys should be aware that detections outside the endpoints of line transects should be discarded to meet the assumptions of line transect DS. Survey configurations with fewer and longer line transects and narrower search widths will require a smaller proportion of detections to be discarded. Retaining these detections in a line transect DS analysis could lead to an overestimation of the detection scale (more detections occur farther from the sampler in point transect than line transect sampling with the same detection scale) and an underestimation of abundance. SCR inherently accounts for this obstacle by explicitly modeling the locations of transect lines and individuals and deriving the distance between them. Therefore, survey effort and detections that are typically discarded in a DS analysis can be incorporated into a SCR analysis without modification of the SCR model.

The SCR model performed well except for the small positive bias of abundance point estimates. However, this bias decreased in scenarios with more recaptures, suggesting that the estimator has negligible bias for large sample sizes [S1 Appendix]. Additionally, the DS abundance estimates from models that accounted for uncertainty in $g(0)$ were as or more positively biased than SCR estimates. The DS abundance estimates from models that fixed $g(0)$ at the true value were unbiased. However, this approach is not ideal in applications of DS, as $g(0)$ is unknown and fixing its value does not propagate any of its associated uncertainty into the abundance estimate [12]. Researchers using these models should be aware of these biases when analyzing small datasets and should consider using the median of the posterior distribution as a point estimate for abundance because it is less biased than the mean [S1 Appendix].

DS has been used to produce spatially explicit estimates of North Atlantic right whale (NARW) abundance [26, 27]; however, SCR should be the preferred approach to estimate NARW abundance based on common NARW sampling procedures and SCR's outperforming DS in our comparisons. NARWs are individually identifiable, and their population is surveyed by aerial line transect surveys conducted over multiple occasions in the same region [27, 29]. Additionally, $g(0) \neq 1$, because NARWs, like other marine mammals, spend time below the water's surface where they are unavailable for detection and may be difficult to detect at the

surface, depending on environmental conditions and behavior [37, 46]. SCR overcomes these sources of bias, while design or modeling modifications would be needed for a DS approach, which would still result in a poorer performing abundance estimator. Furthermore, SCR may produce more robust abundance estimates across competing model formulations than DS and therefore be less sensitive to model misspecification. This is evidenced by both SCR models producing similar abundance estimates (159.0 and 160.2), whereas abundance estimates from the two DS models were so different that their credible intervals did not overlap. The uniform density DS model assumes random placement of transect lines and estimated abundance to be considerably higher than the other three models, which may be due to the non-random placement of transect lines with respect to depth or other environmental characteristics with which NARW abundance is associated. However, we did not apply any model selection techniques or goodness of fit statistics that would be needed to make inference on NARW abundance.

Still, a closed population SCR model is not ideal for estimating NARW abundance. NARWs are a migratory species, and their regional abundance in the SEUS changes throughout the winter as individuals migrate into and out of the region [47]. Therefore, although we estimated NARW abundance in the SEUS over a short 2-week interval to try to meet our SCR model's assumption that the population is closed [7], NARWs may have arrived or departed the SEUS during the study period. Future efforts to estimate NARW abundance in the SEUS using SCR should focus on longer time periods using an open population model, which will allow for estimation of abundance over time and rates of immigration, emigration, and movement within the SEUS region [48–50].

Additional development of the search encounter SCR model may be needed to accommodate NARW social behavior and movement too. SCR assumes individuals in the population of interest are distributed and detected independently, however NARWs form temporary social groups [51]. Future work developing methods to account for group fission/fusion dynamics and the non-independence of detections of individuals in a group could address this violation of SCR assumptions. Although, Bischof et al. [52] found that SCR abundance estimates are robust to non-independence of individuals' activity centers and detections. Furthermore, the SCR models used herein model the space use of a NARW as distributed about a central activity center. This type of space use model is often used for species that, unlike NARWs, exhibit home range behavior. Instead, a movement model could be developed such that the location of a NARW on one sampling occasion is modeled as a function of its location on the previous sampling occasion. Similar models have been developed to describe the movement of individuals' activity centers in open population SCR models [53].

Estimating abundance accurately and precisely is critical to wildlife population management and conservation. Both DS and SCR models estimated abundance with bias of less than 10% across most scenarios. Therefore, it is unlikely that using one model over the other would inform management decisions differently. Still, researchers in the planning and analysis stages of abundance estimation studies should think carefully about their study system when choosing a modeling approach. We advise them to consider additional simulation studies, because the costs of sampling and individual identification and value of accurate and precise abundance estimates vary across study systems. Generally, DS will be more cost efficient than SCR as it does not require repeat sampling occasions or expending effort to identify individuals. The search encounter SCR model requires multiple sampling occasions and the assumption of population closure, which a single occasion DS study does not. However, for rare species such as the NARW, more than one sampling occasion may be required to detect enough individuals for a DS analysis. Still, DS is a good choice when it is safe to assume that g(0) = 1 or perception bias and availability biases can be estimated, when individuals cannot be uniquely identified, and when surveys are conducted over a single sampling occasion. However, researchers must

be aware that a combination of transect line placement and animal movement over multiple occasions can lead to overly precise abundance estimates if variance is inappropriately estimated, which may give management decision-makers false confidence and inaccurate information. Survey design may ameliorate this issue, with surveys conducted in the same region over multiple occasions being distributed widely enough in space or time. This could prevent movement of individuals from causing over-precision in abundance estimates, although surveys should not be so widely distributed in time that abundance changes over sampling occasions. Conversely, search encounter SCR is more flexible and robust than DS for modeling line transect data but requires that individuals are identifiable and surveys are conducted over multiple occasions, which is not possible for every study system. SCR relaxes the DS assumption that detection probability on the transect line is certain without survey design or modeling modifications by using information from individual recaptures across occasions. Additionally, SCR can provide inference on space use and resource selection, which are unaccounted for in DS models and can lead to over-precision and higher error in abundance estimates. Therefore, SCR should be the model of choice to estimate abundance from line transect data collected over multiple occasions on a population of identifiable individuals.

## Supporting information

**S1 Appendix. Supplementary results.**
(DOCX)

**S2 Appendix. Effects of movement on distance sampling abundance estimation.**
(DOCX)

**S3 Appendix. R code.**
(ZIP)

## Acknowledgments

We thank the aerial survey teams that collected the data used to estimate abundance of North Atlantic right whales. We also thank Richard Glennie, Colin Shea, and three anonymous reviewers for providing constructive reviews that improved the manuscript.

## Author Contributions

**Conceptualization:** Nathan J. Crum.

**Formal analysis:** Nathan J. Crum, Lisa C. Neyman, Timothy A. Gowan.

**Methodology:** Nathan J. Crum, Timothy A. Gowan.

**Writing – original draft:** Nathan J. Crum, Lisa C. Neyman.

**Writing – review & editing:** Nathan J. Crum, Lisa C. Neyman, Timothy A. Gowan.

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
