## [Decision Letter · Decision Letter 0]

22 Mar 2021

PONE-D-21-02864

Abundance estimation for line transect sampling: A comparison of distance sampling and spatial capture-recapture models

PLOS ONE

Dear Nathan,

Thank you for submitting your manuscript to PLOS ONE. After careful consideration, we feel that it has merit but does not fully meet PLOS ONE’s publication criteria as it currently stands. Therefore, we invite you to submit a revised version of the manuscript that addresses the points raised during the review process.

Your paper has received three thorough reviews by experts in the field. I and all three reviewers are in agreement that the manuscript represents a great contribution. Due to the thoroughness and thoughtfulness of the reviews, I have no additional comments. Please respond to the comments provided by each of the reviewers.

We look forward to receiving your revised manuscript.

Best,

Angela Fuller

Academic Editor

PLOS ONE

Journal Requirements:

5. Thank you for stating the following after the Acknowledgments Section of your manuscript:

'Funding

North Atlantic right whale aerial surveys were funded by the National Oceanic and Atmospheric Administration, United States Coast Guard, United States Navy, United States Army Corps of Engineers, and Georgia Department of Natural Resources. Analyses of simulation and North Atlantic right whale data were funded by the National Oceanic and Atmospheric Administration.'

'This study was supported by the National Oceanic and Atmospheric Administration (NOAA; grant nos. NA14OAR4170108 and NA16NMF4720319).  The funders had no role in study design, data collection and analysis, decision to publish, or preparation of the manuscript.'

Reviewers' comments:

Reviewer's Responses to Questions

**Comments to the Author**

1. Is the manuscript technically sound, and do the data support the conclusions?

Reviewer #1: Yes

Reviewer #2: Yes

Reviewer #3: Yes

2. Has the statistical analysis been performed appropriately and rigorously? 

Reviewer #1: Yes

Reviewer #2: Yes

Reviewer #3: Yes

3. Have the authors made all data underlying the findings in their manuscript fully available?

Reviewer #1: Yes

Reviewer #2: Yes

Reviewer #3: Yes

4. Is the manuscript presented in an intelligible fashion and written in standard English?

Reviewer #1: Yes

Reviewer #2: Yes

Reviewer #3: Yes

5. Review Comments to the Author

Reviewer #1: Overall comment

The authors compared the performances of distance sampling and spatial capture recapture models to estimate abundance. They performed a simulation study and a real-life case study that allow a complete evaluation of the methods. This work is timely and particularly well implemented. However, I think the manuscript lack of clarity to have a good assessment of the work performed. Apart from raising minor concerns about the methodological aspect, I suggested substantial modifications of the writing and the structure of the manuscript, especially the introduction and the methods sections.

Once the manuscript will be made clearer, I suspect it will be a (very) good paper.

Main comments

Most of my comments target the Introduction and the Methods sections. I had trouble to follow these two paragraphs and I suggested some rearrangements.

Abstract

L13-14. I’m not sure we could say that search encounter SCR have rarely been applied. Doing a quick look at who cited the Royle et al., (2011) paper, I found 40 papers. I have not read all of them but I made sure that several implemented some king of search encounter sampling design. What do you names a search encounter SCR? Do you named “search encounter” SCR a model in which the study area is continuous, rather than composed of discrete sites?

Introduction

To my mind, the first paragraph of the introduction is too short. I would hope to have more contextualization. Why comparing DS and SCR is relevant? What was the results of previous comparisons between DS and classical CR models?

I would not use subtitles in the Introduction, you can keep it neat.

I think there is too much information about modelling techniques in the introduction. I would prefer to present the model in a general context, what do they estimate? from what kind of data? The pros and cons of DS and SCR. What is a search encounter SCR?

I would keep the technical part for the Methods section, which correspond to most of the DS and SCR paragraphs (e.g. modelling assumptions, g(0), the different kinds of modelling techniques, etc).

Most of the “Distance sampling paragraph” in the Introduction should be in the Methods section.

L56: To my mind, you cannot write g(0) for the first time and use it in the entire paragraph without any equation. As you describe a general context, you can remove “g(0)” and discuss in terms of imperfect detection on the transect line.

L57-59: This sentence could be simpler. Maybe something like “Often, detection is not perfect on the transect line due to availability and perception biases. For example, some individuals are unavailable for detection, or observers do not detect all individuals”.  

L72-73: “during a sampling occasion”. And actually, I don’t think it is an issue if an individual is detected multiple times during the same occasion.

The case study paragraph is better suited for the introduction and the DS and SCR.

L124-125: I would remove this sentence.

L127-128: I don’t understand why estimating g(0) and space use parameter would produce less precise estimates for SCR than for DS. Can you explain?

L126-131: There is no relation between these sentences and the Case study subtitle. That is why I would remove subtitle for example.

Methods

I got a bit confused by the organization of the Methods section. I found it hard to follow as there are repetitions with the paragraphs of the Introduction that present DS and SCR. I suggest beginning the Methods by describing the formulation of the DS model and the SCR model. When doing this, I would like to write the explicit formulation of i) the latent abundance model, ii) the detection probability (you added it for the simulation but not for DS or SCR).

Then, you can detail the simulations scenarios. Then, the right whale case study. Finally, the Bayesian implementation.

Simulation scenarios

Why does the number of sampling occasions in not consistent between the different values of g(0) in the simulations?

L151-152: How many occasions?

L163: Why is sigma_d = 1/3?

L181: Why is sigma_m = 4?

DS

You should merge this paragraph with the DS paragraph of the introduction.

It is not clear how you spatialized the DS models to estimates density. I understand you modeled density as a quadratic function of the covariate but I would like to have an explicit formulation of this relationship somewhere.

SCR

Idem, I suggest you merge it with the SCR paragraph of the introduction.

Case study

L259: Why is M=259?

Results

L275-276: I find this sentence hard to follow. Maybe split it in two parts, or add a comma after “0.8”.

Discussion

The discussion is clear and complete. I like it a lot !

One question, how do you explain the difference in abundance estimates between uniform DS and SCR models?

L318: I think some developments exist to deal with such issues. For example, binned detection probability account for imprecision in distance estimation.

Minor comments

General comments about the writing.

In scientific writing, I would reduce the use of “this/these” in the manuscript. I prefer to make shorter sentences with repetitions, which I suspect would make the text clearer without making it heavier. I noticed it in the Introduction and the Methods section, either it disappeared in the lasts sections, or I got used to it while reading…

Keywords: maybe add “simulations”, and “NIMBLE” in the keywords if possible. The first one help to highlight an important part of your work as I suspect your simulations would be looked for by many scientists that use these methods. The second granted the NIMBLE team for their huge work on developing and maintaining NIMBLE.

L13-14: I would remove “this” and repeat “Search encounter spatial capture recapture”. And clarify which models are compared.

L18: I would remove “did”.

L22-23: If you want to make a list, I would repeat “when” at each iteration.

L222-224: I suggest to write the g(0) prior clearer. There is an open bracket that is unclosed. Maybe “as a prior for simulated g(0)= 0.8, we used g(0)~Beta(…, …)”. And repeat every time.

Reviewer #2: Overall I really liked this paper. It is well-written, well organized, and provides a useful comparison among models that should be applicable to real-world data.

I have some overall concerns, which I don't think detract from the value of the info the paper presents, but accounting for them might put the paper into a better context.

First, it's unclear to me how frequently the situation occurs in the real world where you have repeated samples of the same transects within a single short period. Arguably one of the advantages of distance sampling is that you can avoid repeated sampling of the same transects. Are the advantages (in terms of more precise/less biased estimates) worth the extra cost of doing the repeated samples? In this case the authors already had the extra data, but what would be the authors' recommendations to someone designing a new study from the ground up - is the increased modeling flexibility worth the cost (and the added assumptions about individual animal behavior)?

Second, if you have repeated transect data (as the authors do in both the simulations and real world data) wouldn't a distance sampling model specifically designed for this type of data, like Chandler et al. (2011), which allows estimation of temporary emigration/availability, be more appropriate than a standard distance model for comparison? Especially since the authors point out the issue of NARW sometimes being deep underwater and unavailable. The authors cite the Kery and Royle book which discusses this model (section 9.5, pg 483) so perhaps they used it, but it didn't look like it in their methods (or code) to me. Seems like it should at least be mentioned as an alternative.

Specific Comments

L18: What "nominal" credible interval coverage means was not immediately clear to me and probably won't be for many reading the abstract, is there a way to present this more clearly?

L19: You correctly point out the additional SCR assumptions, but maybe also worth pointing out that it avoids another commonly made assumption when using DS, that g(0) = 1? So there are some trade-offs.

INTRO

Overall I thought the summary of the DS/SCR was really well-written and should make this comparison accessible to a wide audience. Nice work.

L44: This is not strictly true, though. For example if animals are avoiding the observer at short distances and there is a "shoulder" at the beginning of the detection curve that you correct for with e.g. a hazard-rate function.

L51: In a way you consider both methods in this study, right - line transect with two points at either end? Or do you mean specifically for the explanation, you only consider line transects?

L61: Double observer/MRDS can't address non-availability (g(0) = 0), though, since presumably neither observer can detect an unavailable animal. Separate studies or maybe multiple repeated transects could allow estimation of availability. Might be worth saying here which reason for g(0) < 1 each analytical approach can potentially address. The way it's worded now it implies either approach could solve either (or both) reasons for g(0) < 1. Or maybe phrase it as MRDS/double observer can handle 0 < g(0) < 1, while a secondary study or perhaps repeated samples (e.g., Chandler et al. (2011) "Inference about density and temporary emigration in unmarked populations") could handle g(0) = 0 during a given sampling event.

L65: See also Oyster et al. (2018) "Hierarchical mark‐recapture distance sampling to estimate moose abundance" for an interesting recent approach to estimating availability to aerial transects.

L73: "at most once during sampling"  "at most once during a sampling *occasion*. I tend to think of "sampling" as all the sampling occasions combined.

L120: "NARWs often move between sampling occasions" - is this a reason DS is less appropriate? Presumably with DS you would normally be sampling a given transect only once (I think your protocol is a bit unusual in this regard?) so this movement wouldn't be relevant. Unless you think they are being double-counted on different transects that occur at different times.

L147: Could emphasize here that your range of g(0) values does not include a scenario where some animals were actually completely unavailable to be detected.

L151: Was the number of occasions per scenario always > 1? I assume so unless there were some scenarios where you only fit DS models. This begs the question, to me: if you have repeated transects already, wouldn't a generalized distance sampling model like Chandler et al. 2011 that uses repeated samples of a transect/point to estimate temporary emigration/availability in addition to abundance be an interesting comparison to make with SCR?

It's a bit unclear to me how you handled repeated samples of the same transect with DS. Did you just treat each transect * occasion as an independent "site"?

Also, to what extent do NARW fit the general assumption of SCR, e.g. that individuals are moving around an unobserved home range center? I don't know much about whales but it seems less likely to be the case than e.g. with small mammals, or birds moving around a territory (as in the original Royle paper).

METHODS

Paragraph beginning L155: Easy here to get confused between distance to activity center (d_ij) and distance between individual to transect (distance_ij) and the associated scale parameters. Maybe add a summary sentence distinguishing between the two? Or use a different term/parameter name for 'distance_ij' (or vice-versa) so it is not so easy to get confused.

L201-202: How frequent were these situations where you had to discard a simulated dataset due to non-convergence? Did it happen at the same rate for DS and SCR models? Any ideas as to what characteristics of the datasets made them fail to converge? This seems like useful practical information - I would guess that for a given dataset you would be more likely to have convergence problems with the SCR model given its additional complexity. If so that is a potential additional drawback of SCR since presumably some real-world datasets will also have convergence problems.

L190: Might want to mention that the models are described in detail in the next section, I got a bit confused by the organization here.

L205: Might want a short paranthetical explaining what you mean by "credible interval coverage".

L211: This combo of point and line data collection is pretty interesting and your approach to solving it makes sense to me. However, I wonder how often this end-of-line data is collected in typical DS sampling datasets? What percent of your real-world observations fell into these points at the end? Did you check to see if your estimates from DS and SCR were any different if they were excluded? The point + line thing is just an additional layer of complexity here that might be better to minimize in a comparison paper like this if it's not a situation that is likely to be generally true in other studies.

L240-242: This data definitely seems like a perfect candidate for the generalized distance sampling model I mentioned earlier (e.g. see gdistsamp() in R package unmarked) especially since you don't appear to have double observer data. Might be a more "fair" comparison with SCR since both would make full use of the repeated samples. At least worth mentioning that it could be done.

RESULTS

L283: I'm not exactly sure what you mean by nominal/not significantly different from 95% here. Do you mean that 95% of the 95% credible intervals contained the true value? Might explain this a bit more.

DISCUSSION

L314-316: Use of the SCR works under the assumption that transects were surveyed multiple times within a short time frame. My sense is that part of the appeal of distance sampling is only having to survey once. Are the advantages of the SCR worth the extra cost of multiple surveys (either in $ or by reducing the total number of sites surveyed) over just doing DS on one transect? Would be interesting to see what kind of abundance estimates single-occasion DS (either on simulated data or your NARW data) gave relative to SCR.

L338: Repeating earlier comment: I wonder how often data are actually collected at the end of the transect lines like this. Definitely something to be careful of.

L389: I tend to agree that the assumption of an animal moving around an activity center does not seem to match NARW. How do you think this might have affected your abundance estimates, if at all?

L416-417: Again, what about generalized distance sampling estimating availablity (vs. standard DS)? I would still expect SCR to be better, but an advantage of GDS vs SCR is that you wouldn't have to uniquely ID individuals.

Reviewer #3: This paper compares hierarchical (and 2-stage) distance sampling to spatial capture-recapture using a set of simulations, and then applies both models to a breeding season survey of North Atlantic right whales. The authors find that SCR models outperform the DS models, particularly in the presence of unmodeled spatial variation in density. The paper is well written and the analyses well executed. This paper is very useful (the NIMBLE code is great) and will receive properly warranted attention. I have only a few thoughts.

The movement model used to simulate individuals on the landscape is very basic and meets the assumptions for SCR while potentially violating some assumptions for DS. This could be emphasized a little more, though the latter point is definitely explored and discussed. Comparing two estimators that vary in assumption violations may seem a little unfair. Still, the right whale survey design dictates the need for this narrative (SCR vs. DS) and the authors do caution (L335) that only a single space use model was considered.

Along those lines, the closure assumption would be a problem for the SCR model but was not explored in simulations. Not saying that it needs to be, but this is where short survey durations give DS models the advantage. Getting enough spatial recaptures is the challenge for SCR models so there is a tension between closure violation and survey duration. Either way, the authors do discuss the need for more complicated SCR models to accommodate right whales and I agree with this. Individual heterogeneity is another potential problem, particularly if females with calves are more visible (e.g., staying near surface), though technically this could affect the DS models as well.

It was interesting that spatial variation as determined by a quadratic function induced bias in the constant-density SCR model. The DS model assumes that transects are distributed randomly with regards to spatial variation, so violating this assumption would be expected to result in poor performance. But one advantage of SCR models that is often cited is that the homogeneous point process works well despite unmodeled spatial variation. The differences in estimates for right whales suggests something slightly different may be at play.

My final comment is that Table 1 would be more compelling as a figure, especially in the main text. Not saying this needs to be done! Many simulation papers present these results in tables, but some (e.g., Feinberg & Wainer 2011) would argue the messages are better presented visually. Some ggplot code would make quick work of these results. Just something to consider.

Feinberg, R. A., and H. Wainer. "Extracting sunbeams from cucumbers." Journal of Computational and Graphical Statistics 20.4 (2011): 793-810.

Dan Linden

daniel.linden@noaa.gov

L90 Better to just state that SCR models can accommodate other survey methods, rather than referring to “advances”.

L157 “some region u” is a bit confusing. I would state that “u” represents an area of continuous space, or a collection of discrete pixels at some resolution.

L176 The values of B0/B1 to generate N=150 depend on the distribution of the simulated covariate values. Might be better to just state that the values were chosen to generate E(N)=150 and interested folks can find the exact values in the appendix code (along with the spatial covariate). Or make it clear what the covariate_n distribution was.

L207 The theta subscript here looks like an “l” and not an “i”.

L220-226 The alpha/beta values for the Beta distributions could be thrown into a small appendix table. They are a little distracting here and simply indicate the uncertainty induced for each g0 scenario.

L239 Can you translate these UTMs to rough latitudes? The map makes the location pretty clear and the UTM coordinates are not otherwise useful to most readers.

L307 Bias, RMSE, and coverage are all about accuracy – better to succinctly state that you compared the accuracy, especially for an intro discussion paragraph.

6. PLOS authors have the option to publish the peer review history of their article (what does this mean?). If published, this will include your full peer review and any attached files.

Reviewer #1: No

Reviewer #2: No

Reviewer #3: No

---

## [Author Response · Author response to Decision Letter 0]

27 Apr 2021

Dear Dr. Fuller,

We thank you for the opportunity to submit a revised manuscript for consideration to be published in PLOS ONE. Please see below for a point by point response to the reviewers comments.

Reviewer #1: Overall comment

The authors compared the performances of distance sampling and spatial capture recapture models to estimate abundance. They performed a simulation study and a real-life case study that allow a complete evaluation of the methods. This work is timely and particularly well implemented. However, I think the manuscript lack of clarity to have a good assessment of the work performed. Apart from raising minor concerns about the methodological aspect, I suggested substantial modifications of the writing and the structure of the manuscript, especially the introduction and the methods sections.

Once the manuscript will be made clearer, I suspect it will be a (very) good paper.

Main comments

Most of my comments target the Introduction and the Methods sections. I had trouble to follow these two paragraphs and I suggested some rearrangements.

Abstract

L13-14. I’m not sure we could say that search encounter SCR have rarely been applied. Doing a quick look at who cited the Royle et al., (2011) paper, I found 40 papers. I have not read all of them but I made sure that several implemented some king of search encounter sampling design. What do you names a search encounter SCR? Do you named “search encounter” SCR a model in which the study area is continuous, rather than composed of discrete sites?

We now emphasize in the introduction (L108-109) that the search encounter SCR model differs from traditional SCR models in that individuals can be detected anywhere within the state space and not just at traps. It is true that many studies have used search encounter field methods, but these studies used traditional SCR models, i.e., they aggregated survey effort and the location of detections into “trapping grids” as we mention in L108-113. Here we are referring to the model and not the sampling design. 

Introduction

To my mind, the first paragraph of the introduction is too short. I would hope to have more contextualization. Why comparing DS and SCR is relevant? What was the results of previous comparisons between DS and classical CR models?

We feel that L51-57 concisely addresses the importance of comparing DS and SCR. Our goal is to point out that SCR and DS are commonly used abundance estimators, but that it is unclear which performs better in situations where both could be used. 

I would not use subtitles in the Introduction, you can keep it neat.

We have removed subtitles from the introduction.

I think there is too much information about modelling techniques in the introduction. I would prefer to present the model in a general context, what do they estimate? from what kind of data? The pros and cons of DS and SCR. What is a search encounter SCR?

I would keep the technical part for the Methods section, which correspond to most of the DS and SCR paragraphs (e.g. modelling assumptions, g(0), the different kinds of modelling techniques, etc).

Most of the “Distance sampling paragraph” in the Introduction should be in the Methods section.

We have removed obvious statistical jargon from the introduction, including references to g(0). However, we contend that this manuscript’s scope is the performance of two statistical models and that it is important for the reader to be aware of the concepts, including model assumptions and techniques, that are covered in the introduction. Those assumptions and techniques provide important context for considering the pros and cons of DS and SCR. Therefore, we have not moved this content to the Methods section. 

L56: To my mind, you cannot write g(0) for the first time and use it in the entire paragraph without any equation. As you describe a general context, you can remove “g(0)” and discuss in terms of imperfect detection on the transect line.

We have now removed reference to g(0) from the introduction.

L57-59: This sentence could be simpler. Maybe something like “Often, detection is not perfect on the transect line due to availability and perception biases. For example, some individuals are unavailable for detection, or observers do not detect all individuals”. 

We have split this sentence in a similar fashion following this suggestion (L74-77).

L72-73: “during a sampling occasion”. And actually, I don’t think it is an issue if an individual is detected multiple times during the same occasion.

Traditional SCR can permit multiple detections of an individual on one occasion with a Poisson detection model, for instance, but the search encounter SCR model used in this manuscript does not permit more than one detection during a sampling occasion, hence our focus on such cases. We refer to the need for multiple occasions for the SCR model throughout the manuscript, e.g., L120-122 and L518-519.

The case study paragraph is better suited for the introduction and the DS and SCR.

L124-125: I would remove this sentence.

SCR being able to estimate g(0) without additional data is an important feature and difference from DS. Therefore, we feel leaving this sentence in the manuscript is warranted (now L137-139).

L127-128: I don’t understand why estimating g(0) and space use parameter would produce less precise estimates for SCR than for DS. Can you explain?

In short, parameter estimates have higher variance and lower bias as a model becomes more complex, i.e., as parameters are added to the model. However, SCR and DS do not have the same model structure, so we removed this statement to avoid any confusion.

L126-131: There is no relation between these sentences and the Case study subtitle. That is why I would remove subtitle for example.

We have removed the subtitles from the introduction section.

Methods

I got a bit confused by the organization of the Methods section. I found it hard to follow as there are repetitions with the paragraphs of the Introduction that present DS and SCR. I suggest beginning the Methods by describing the formulation of the DS model and the SCR model. When doing this, I would like to write the explicit formulation of i) the latent abundance model, ii) the detection probability (you added it for the simulation but not for DS or SCR).

Then, you can detail the simulations scenarios. Then, the right whale case study. Finally, the Bayesian implementation.

We have completely revised the DS and SCR subsections to now include formulations for the models. Additionally, we moved these sections ahead of the Analysis of simulated datasets section. We feel that the Simulation scenarios section provides needed context and ties together the modeling sections. For instance, it introduces that we consider scenarios where density is constant and where expected density is a function of a spatial covariate, which require different model formulations. Therefore, we left the Simulation scenarios section at the beginning of the Methods section. 

Simulation scenarios

Why does the number of sampling occasions in not consistent between the different values of g(0) in the simulations?

We explicitly state here (L167-168) that the number of sampling occasions varies so that the number of detections across scenarios is similar. 

L151-152: How many occasions?

Table 1 (L364) lists the number of occasions used for each scenario.

L163: Why is sigma_d = 1/3?

L181: Why is sigma_m = 4?

We now include an explanation justifying our choice of parameter values (L197-199). The parameters and sampling design were chosen such that an individual with an activity center on the edge of the state space would have a negligible probability of being detected over the course of the study.

DS

You should merge this paragraph with the DS paragraph of the introduction.

It is not clear how you spatialized the DS models to estimates density. I understand you modeled density as a quadratic function of the covariate but I would like to have an explicit formulation of this relationship somewhere.

We have completely revised this section to explicitly describe the model formulations for both the uniform density and spatial covariate models (L224-268).

SCR

Idem, I suggest you merge it with the SCR paragraph of the introduction.

Likewise, we have revised this section to describe model formulations more thoroughly (L269-298).

Case study

L259: Why is M=259?

We have edited this to show that 259 unobserved encounter histories were added through data augmentation to arrive at M = 350. We have added an explanation for this choice at L350-352. The posterior distribution of abundance was not truncated by our selection of M=350, which can cause problems when the selected M is too small.

Results

L275-276: I find this sentence hard to follow. Maybe split it in two parts, or add a comma after “0.8”.

We have split this sentence into two sentences (L372-374).

Discussion

The discussion is clear and complete. I like it a lot !

One question, how do you explain the difference in abundance estimates between uniform DS and SCR models?

We have added a potential explanation as to why the uniform density DS model estimated abundance so much higher than the other models (L484-487).

L318: I think some developments exist to deal with such issues. For example, binned detection probability account for imprecision in distance estimation.

In the case of binned distances, there is still an assumption that detections are not incorrectly binned. This is the same concept but with discrete rather than continuous distances (L418-420).

Minor comments

General comments about the writing.

In scientific writing, I would reduce the use of “this/these” in the manuscript. I prefer to make shorter sentences with repetitions, which I suspect would make the text clearer without making it heavier. I noticed it in the Introduction and the Methods section, either it disappeared in the lasts sections, or I got used to it while reading…

We have removed “this”/”these” throughout the introduction where they are not essential.

Keywords: maybe add “simulations”, and “NIMBLE” in the keywords if possible. The first one help to highlight an important part of your work as I suspect your simulations would be looked for by many scientists that use these methods. The second granted the NIMBLE team for their huge work on developing and maintaining NIMBLE.

We agree that NIMBLE facilitated this work and therefore now include it as a keyword (L48).

L13-14: I would remove “this” and repeat “Search encounter spatial capture recapture”. And clarify which models are compared.

We have made this change (L30-31).

L18: I would remove “did”.

We have made this change (L34-35).

L22-23: If you want to make a list, I would repeat “when” at each iteration.

We have made this change (L40-44).

L222-224: I suggest to write the g(0) prior clearer. There is an open bracket that is unclosed. Maybe “as a prior for simulated g(0)= 0.8, we used g(0)~Beta(…, …)”. And repeat every time.

In revising the DS subsection of the methods, we removed this sentence.

Reviewer #2: Overall I really liked this paper. It is well-written, well organized, and provides a useful comparison among models that should be applicable to real-world data.

I have some overall concerns, which I don't think detract from the value of the info the paper presents, but accounting for them might put the paper into a better context.

First, it's unclear to me how frequently the situation occurs in the real world where you have repeated samples of the same transects within a single short period. Arguably one of the advantages of distance sampling is that you can avoid repeated sampling of the same transects. Are the advantages (in terms of more precise/less biased estimates) worth the extra cost of doing the repeated samples? In this case the authors already had the extra data, but what would be the authors' recommendations to someone designing a new study from the ground up - is the increased modeling flexibility worth the cost (and the added assumptions about individual animal behavior)?

Answers to these questions are very context dependent. Cost of sampling and individual identification will vary across study systems, as will the value of accuracy and precision in abundance estimates. We added to the discussion (L514-516) recommending that if you are designing a new study you should simulate data of your own that are representative of your study system. Additionally, the last paragraph of the discussion details general recommendations (L510-538). Also, we point out that although we make assumptions about individual space use in the SCR model, not modeling individual space use, as in DS, also led to the issue of underestimating variance in abundance, one of our principal results (L421-446).

Second, if you have repeated transect data (as the authors do in both the simulations and real world data) wouldn't a distance sampling model specifically designed for this type of data, like Chandler et al. (2011), which allows estimation of temporary emigration/availability, be more appropriate than a standard distance model for comparison? Especially since the authors point out the issue of NARW sometimes being deep underwater and unavailable. The authors cite the Kery and Royle book which discusses this model (section 9.5, pg 483) so perhaps they used it, but it didn't look like it in their methods (or code) to me. Seems like it should at least be mentioned as an alternative.

The Chandler et al. (2011) model, and similar models that would allow for modeling the effects of spatial covariates on density, e.g., Mizel et al. (2017), estimate superpopulation size and temporary emigration. Additionally, density is derived as the product of the superpopulation size and 1 minus the temporary emigration rate. However, the model assumes that g(0) = 1 for individuals within the sampled region. So, if g(0) is less than 1, these models will underestimate density. We analyzed 100 simulated datasets using the Chandler et al. model under each of our single observer uniform density scenarios with a population size of 100 individuals and g(0) equals 0.8, 0.5, and 0.3 (R code provided in supporting information). We estimated abundance as the product of density and the area of the state space. We note that the superpopulation and temporary emigration rates were estimated very imprecisely but were strongly negatively correlated, hence density and abundance were estimated with greater precision. We provide a table summarizing the results below, and as you can see, density is underestimated at a rate proportional to the value of g(0).

# of Occasions g(0) Abundance Mean Abundance 95% Credible Interval Superpopulation Mean Superpopulation 95% Credible Interval Temporary Emigration Mean Temporary Emigration 95% Credible Interval

20 0.3 30.99 (22.87-40.64) 90.60 (20.51-538.81) 0.53 (0.06-0.97)

10 0.5 49.67 (35.45-66.88) 61.05 (12.91-376.90) 0.49 (0.05-0.96)

6 0.8 83.47 (59.54-112.55) 50.88 (8.73-363.15) 0.47 (0.04-0.92)

We suspect that the Chandler et al. (2011) model could be used when g(0) is less than one if it is safe to assume that the sampled region is demographically closed across sampling occasions. This is not a safe assumption in the case of our North Atlantic right whale case study or simulations, where both temporary emigration and availability processes affect whether individuals are detected or not. We suspect this may not be a safe assumption in many other real world applications where g(0) is less than 1. Nevertheless, in such cases, 1 minus the temporary emigration rate could be interpreted as g(0) and the superpopulation size could be estimated as abundance in the sampled region (assuming demographic closure within that region). However, as we mentioned above, these parameters were estimated very imprecisely and were strongly negatively correlated in our simulation analyses. 

We now cite Chandler et al. (2011) and mention that DS models have been developed to account for temporary emigration from the sampled region (L428-432). Based on these results and our understanding of the Chandler et al. (2011) model, e.g., Chandler et al. state that variance is not appropriately estimated when the probability of temporary emigration differs across individuals on page 1433, we do not think the model warrants further evaluation.

Specific Comments

L18: What "nominal" credible interval coverage means was not immediately clear to me and probably won't be for many reading the abstract, is there a way to present this more clearly?

We now provide a brief description of what is meant by nominal (L36-37).

L19: You correctly point out the additional SCR assumptions, but maybe also worth pointing out that it avoids another commonly made assumption when using DS, that g(0) = 1? So there are some trade-offs.

We have added that SCR should be considered when there is imperfect detection along the transect line (L42-43).

INTRO

Overall I thought the summary of the DS/SCR was really well-written and should make this comparison accessible to a wide audience. Nice work.

L44: This is not strictly true, though. For example if animals are avoiding the observer at short distances and there is a "shoulder" at the beginning of the detection curve that you correct for with e.g. a hazard-rate function.

This reference to modeling detection probability as a decreasing function of distance has been removed.

L51: In a way you consider both methods in this study, right - line transect with two points at either end? Or do you mean specifically for the explanation, you only consider line transects?

This is in reference to survey design. We have changed line transect method to line transect surveys to emphasize this point (L70). 

L61: Double observer/MRDS can't address non-availability (g(0) = 0), though, since presumably neither observer can detect an unavailable animal. Separate studies or maybe multiple repeated transects could allow estimation of availability. Might be worth saying here which reason for g(0) < 1 each analytical approach can potentially address. The way it's worded now it implies either approach could solve either (or both) reasons for g(0) < 1. Or maybe phrase it as MRDS/double observer can handle 0 < g(0) < 1, while a secondary study or perhaps repeated samples (e.g., Chandler et al. (2011) "Inference about density and temporary emigration in unmarked populations") could handle g(0) = 0 during a given sampling event.

We now explicitly state that mark recapture distance sampling can be used to account for perception bias (L79-81). As we mentioned in our above response to the second main comment, the Chandler et al. (2011) assumes that g(0) = 1 unless the sampled region is demographically closed. And even if this assumption is met, the superpopulation estimate, which we found to be very imprecise, would be interpreted as the abundance estimate. Therefore, the Chandler model is not a good fit for the data described in the manuscript. 

L65: See also Oyster et al. (2018) "Hierarchical mark‐recapture distance sampling to estimate moose abundance" for an interesting recent approach to estimating availability to aerial transects.

We now cite this article as a way to address availability bias (L83-84).

L73: "at most once during sampling"  "at most once during a sampling *occasion*. I tend to think of "sampling" as all the sampling occasions combined.

We have made this change for clarity (L89-90).

L120: "NARWs often move between sampling occasions" - is this a reason DS is less appropriate? Presumably with DS you would normally be sampling a given transect only once (I think your protocol is a bit unusual in this regard?) so this movement wouldn't be relevant. Unless you think they are being double-counted on different transects that occur at different times.

We now point out that the SCR model may be more appropriate, conceptually, because inference about NARW movement may be of interest, not because movement is a problem for DS (L139-141).

L147: Could emphasize here that your range of g(0) values does not include a scenario where some animals were actually completely unavailable to be detected.

We argue that this point is not completely accurate or straightforward, therefore we do not mention it. Although we did not include an availability component in the simulations, e.g. draw from a Bernoulli distribution to determine whether an individual was available or not, there were large areas of the state space with effectively 0 probability of detection. Therefore, some individuals were unavailable for detection based on their movement outcomes. Additionally, single observer DS cannot distinguish or estimate perception and availability bias for individuals within the sampled region, and single observer SCR estimates g(0) as the product of availability and perception. So, our single observer simulations are functionally equivalent to having a scenario where the product of availability and perception equals our g(0) values, meaning availability could have be less than 1. 

L151: Was the number of occasions per scenario always > 1? I assume so unless there were some scenarios where you only fit DS models. This begs the question, to me: if you have repeated transects already, wouldn't a generalized distance sampling model like Chandler et al. 2011 that uses repeated samples of a transect/point to estimate temporary emigration/availability in addition to abundance be an interesting comparison to make with SCR?

Yes, we always refer to occasions, plural, and list the number of occasions for each scenario in Table 1 (L364). Regarding the Chandler et al. 2011 model, please refer to our comment above in response to the second main concern.

It's a bit unclear to me how you handled repeated samples of the same transect with DS. Did you just treat each transect * occasion as an independent "site"?

Detection/non-detection events are considered independent in this Bayesian formulation. We have revised the DS modeling subsection of the Methods in a way that should make this evident (L224-268). Additionally, we discuss different approaches to variance estimation in the Discussion (L422-441) and in S2 Appendix.

Also, to what extent do NARW fit the general assumption of SCR, e.g. that individuals are moving around an unobserved home range center? I don't know much about whales but it seems less likely to be the case than e.g. with small mammals, or birds moving around a territory (as in the original Royle paper).

We mention that our space use models may not be realistic in the discussion (L441-446, L504-509). We agree that the activity center model is not ideal for the right whale application. Although, it does allow us to model the effects of spatial covariates, e.g., depth, on movement outcomes. Developing movement models that better fit the right whale movement process is outside the scope of this study, but it is a topic on which we are actively working for future analyses.

METHODS

Paragraph beginning L155: Easy here to get confused between distance to activity center (d_ij) and distance between individual to transect (distance_ij) and the associated scale parameters. Maybe add a summary sentence distinguishing between the two? Or use a different term/parameter name for 'distance_ij' (or vice-versa) so it is not so easy to get confused.

We have changed these to dmovement,i,j and ddetection,i,j to better distinguish the two distance variables (L188, L195).

L201-202: How frequent were these situations where you had to discard a simulated dataset due to non-convergence? Did it happen at the same rate for DS and SCR models? Any ideas as to what characteristics of the datasets made them fail to converge? This seems like useful practical information - I would guess that for a given dataset you would be more likely to have convergence problems with the SCR model given its additional complexity. If so that is a potential additional drawback of SCR since presumably some real-world datasets will also have convergence problems.

We have added these results (L387-393). Convergence was not a problem outside of the spatial covariate scenario and the scenario where g(0)=0.3. But in both cases, SCR models converged more frequently than DS models that accounted for uncertainty in g(0).

L190: Might want to mention that the models are described in detail in the next section, I got a bit confused by the organization here.

We revised the methods section and placed the model description subsections ahead of the analysis subsection.

L205: Might want a short paranthetical explaining what you mean by "credible interval coverage".

We added text and an equation to provide a description of coverage (L312-316).

L211: This combo of point and line data collection is pretty interesting and your approach to solving it makes sense to me. However, I wonder how often this end-of-line data is collected in typical DS sampling datasets? What percent of your real-world observations fell into these points at the end? Did you check to see if your estimates from DS and SCR were any different if they were excluded? The point + line thing is just an additional layer of complexity here that might be better to minimize in a comparison paper like this if it's not a situation that is likely to be generally true in other studies.

We have minimized our explanation of this concept here and point interested readers to S3 Appendix to see how this is done (L252-258). We include this point in the manuscript to point out that the SCR model naturally accommodates data that have to be discarded or modeled separately in a DS model.

L240-242: This data definitely seems like a perfect candidate for the generalized distance sampling model I mentioned earlier (e.g. see gdistsamp() in R package unmarked) especially since you don't appear to have double observer data. Might be a more "fair" comparison with SCR since both would make full use of the repeated samples. At least worth mentioning that it could be done.

Please refer to our response to the second main concern listed above. 

RESULTS

L283: I'm not exactly sure what you mean by nominal/not significantly different from 95% here. Do you mean that 95% of the 95% credible intervals contained the true value? Might explain this a bit more.

We have added an explanation of coverage to the analysis of simulated datasets section (L312-316).

DISCUSSION

L314-316: Use of the SCR works under the assumption that transects were surveyed multiple times within a short time frame. My sense is that part of the appeal of distance sampling is only having to survey once. Are the advantages of the SCR worth the extra cost of multiple surveys (either in $ or by reducing the total number of sites surveyed) over just doing DS on one transect? Would be interesting to see what kind of abundance estimates single-occasion DS (either on simulated data or your NARW data) gave relative to SCR.

We address some of this comment in our response to the reviewer’s first main concern. Additionally, we do not think it would be fair to compare DS estimates from a single sampling occasion to SCR estimates. In simulated datasets the number of detections per sampling occasion was almost always under 10. It would not be advisable to estimate abundance based on such a small sample in a real world application.

L338: Repeating earlier comment: I wonder how often data are actually collected at the end of the transect lines like this. Definitely something to be careful of.

Again, we include this point to emphasize that there are no complications with incorporating these data into a SCR model, whereas there are with DS. 

L389: I tend to agree that the assumption of an animal moving around an activity center does not seem to match NARW. How do you think this might have affected your abundance estimates, if at all?

We are unsure how this might have affected abundance estimates. Although, our SCR analysis that modeled movement outcomes and activity center density as functions of depth estimated abundance similarly to the model without the depth covariate. Testing different movement models for NARW SCR models is beyond the scope of this study and is something we are actively working on. 

L416-417: Again, what about generalized distance sampling estimating availablity (vs. standard DS)? I would still expect SCR to be better, but an advantage of GDS vs SCR is that you wouldn't have to uniquely ID individuals.

Please refer to our response to this comment above.

Reviewer #3: This paper compares hierarchical (and 2-stage) distance sampling to spatial capture-recapture using a set of simulations, and then applies both models to a breeding season survey of North Atlantic right whales. The authors find that SCR models outperform the DS models, particularly in the presence of unmodeled spatial variation in density. The paper is well written and the analyses well executed. This paper is very useful (the NIMBLE code is great) and will receive properly warranted attention. I have only a few thoughts.

The movement model used to simulate individuals on the landscape is very basic and meets the assumptions for SCR while potentially violating some assumptions for DS. This could be emphasized a little more, though the latter point is definitely explored and discussed. Comparing two estimators that vary in assumption violations may seem a little unfair. Still, the right whale survey design dictates the need for this narrative (SCR vs. DS) and the authors do caution (L335) that only a single space use model was considered.

We have added to the discussion to point out that we did not evaluate whether SCR is robust to misspecification of the space use model (443-446). 

Along those lines, the closure assumption would be a problem for the SCR model but was not explored in simulations. Not saying that it needs to be, but this is where short survey durations give DS models the advantage. Getting enough spatial recaptures is the challenge for SCR models so there is a tension between closure violation and survey duration. Either way, the authors do discuss the need for more complicated SCR models to accommodate right whales and I agree with this. Individual heterogeneity is another potential problem, particularly if females with calves are more visible (e.g., staying near surface), though technically this could affect the DS models as well.

We have added to the discussion to emphasize that SCR requires the population closure assumption (L518-519), although testing this was not within the scope of this study. However, Gowan et al. (forthcoming) present an open population version of this SCR model that also accounts for group effects (e.g., different detectability and movement for calving females than others).

It was interesting that spatial variation as determined by a quadratic function induced bias in the constant-density SCR model. The DS model assumes that transects are distributed randomly with regards to spatial variation, so violating this assumption would be expected to result in poor performance. But one advantage of SCR models that is often cited is that the homogeneous point process works well despite unmodeled spatial variation. The differences in estimates for right whales suggests something slightly different may be at play.

Unfortunately, we only saved the posterior distributions for abundance in the misspecification scenario presented in S1 Appendix, Table S1.5. So, we would need to re-run this analysis to determine if this bias is associated with any other parameter biases in the model. In the scenario where density was simulated as a quadratic function of the covariate, expected density was highest in the center of the state space where sampling occurred. Additionally, space use was modeled as a quadratic function of covariates too. Perhaps, the combination of the transect lines covering the high density area and the space use model pushing the few individuals with activity centers outside the survey area to have movement outcomes in the survey area resulted in the positive bias. In the right whale analysis, transect lines covered a greater range of depths in the study area in comparison to the covariate values covered by transect lines in the simulations.

My final comment is that Table 1 would be more compelling as a figure, especially in the main text. Not saying this needs to be done! Many simulation papers present these results in tables, but some (e.g., Feinberg & Wainer 2011) would argue the messages are better presented visually. Some ggplot code would make quick work of these results. Just something to consider.

Feinberg, R. A., and H. Wainer. "Extracting sunbeams from cucumbers." Journal of Computational and Graphical Statistics 20.4 (2011): 793-810.

We have decided to leave Table 1 as it is. We feel that the table is more concise, as we would need multiple figures to represent all of the information presented in the table.

L90 Better to just state that SCR models can accommodate other survey methods, rather than referring to “advances”.

We have made this change (L106).

L157 “some region u” is a bit confusing. I would state that “u” represents an area of continuous space, or a collection of discrete pixels at some resolution.

We have made this change (L185-190).

L176 The values of B0/B1 to generate N=150 depend on the distribution of the simulated covariate values. Might be better to just state that the values were chosen to generate E(N)=150 and interested folks can find the exact values in the appendix code (along with the spatial covariate). Or make it clear what the covariate_n distribution was.

We have removed the values for beta_0 and beta_1. We now only mention that their values were selected such that the expected population size is 150 (L211-212).

L207 The theta subscript here looks like an “l” and not an “i”.

We have changed the subscripts to appear as lower case “i” now (L313-314).

L220-226 The alpha/beta values for the Beta distributions could be thrown into a small appendix table. They are a little distracting here and simply indicate the uncertainty induced for each g0 scenario.

We have removed these values from the main text.

L239 Can you translate these UTMs to rough latitudes? The map makes the location pretty clear and the UTM coordinates are not otherwise useful to most readers.

We now present latitudes rather than UTM coordinates (L323-326).

L307 Bias, RMSE, and coverage are all about accuracy – better to succinctly state that you compared the accuracy, especially for an intro discussion paragraph.

We have changed this to “accuracy and precision” (L409).

References

Chandler, RB, Royle, JA, King, DI. Inference about density and temporary emigration in unmarked populations. Ecology 2011;92:1249-1435.

Gowan, TA, Crum, NC, Roberts, JJ. An open spatial capture–recapture model for estimating density, movement, and population dynamics from line-transect surveys. Ecology and Evolution; forthcoming.

Mizel, JD, Schmidt, JH, Lindberg, MS. Accommodating temporary emigration in spatial distance sampling models. Journal of Applied Ecology 2017;55:1456-1464.

---

## [Editor Report · Decision Letter 1]

12 May 2021

Abundance estimation for line transect sampling: A comparison of distance sampling and spatial capture-recapture models

PONE-D-21-02864R1

Dear Mr. Crum,

Thank you for your attention to the comments and suggestions provided by the three reviewers. We’re pleased to inform you that your manuscript has been judged scientifically suitable for publication and will be formally accepted for publication once it meets all outstanding technical requirements.

Regards,

Angela Fuller

Academic Editor

PLOS ONE

---

## [Editor Report · Acceptance letter]

17 May 2021

PONE-D-21-02864R1 

 Abundance estimation for line transect sampling: A comparison of distance sampling and spatial capture-recapture models 

Dear Dr. Crum:

I'm pleased to inform you that your manuscript has been deemed suitable for publication in PLOS ONE. Congratulations! Your manuscript is now with our production department. 

Kind regards, 

on behalf of

Dr. Angela K. Fuller 

Academic Editor

PLOS ONE